# PeerJ

# Effects of low and high dose intraarticular tiludronate on synovial fluid and clinical variables in healthy horses—a preliminary investigation

Katja F. Duesterdieck-Zellmer[1], Lindsey Moneta[2], Jesse F. Ott[1], Maureen K. Larson[1], Elena M. Gorman[3], Barbara Hunter[1], Christiane V. Löhr[3], Mark E. Payton[4], Jeffrey T. Morré[5] and Claudia S. Maier[5]

[1] Department of Clinical Sciences, College of Veterinary Medicine, Oregon State University, Corvallis, OR, USA
[2] College of Veterinary Medicine, Oregon State University, Corvallis, OR, USA
[3] Department of Biomedical Sciences, College of Veterinary Medicine, Oregon State University, Corvallis, OR, USA
[4] Department of Statistics, Oklahoma State University, Stillwater, OK, USA
[5] Department of Chemistry, Oregon State University, Corvallis, OR, USA

Corresponding author
Katja F. Duesterdieck-Zellmer,
katja.zellmer@oregonstate.edu

## ABSTRACT

To determine effects of intraarticularly administered tiludronate on articular cartilage *in vivo*, eight healthy horses were injected once with tiludronate (low dose tiludronate [LDT] 0.017 mg, $n = 4$; high dose tiludronate [HDT] 50 mg, $n = 4$) into one middle carpal joint and with saline into the contralateral joint. Arthrocentesis of both middle carpal joints was performed pre-treatment, and 10 min, 24 h, 48 h, 7 and 14 days after treatment. Synovial nucleated cell counts and total solids, tiludronate, sulfated glycosaminoglycan (sGAG), chondroitin sulfate 846 epitope (CS-846, a measure of aggrecan synthesis), and collagen type II cleavage neoepitope (C2C) concentrations were determined. Histologic analysis of joint tissues and sGAG quantitation in cartilage was performed at 14 days in HDT horses. Data were analyzed by repeated measures non-parametric ANOVA and Wilcoxon signed-rank test. High dose tiludronate administration produced synovial fluid tiludronate concentrations of 2,677,500 ng/mL, exceeding concentrations that were safe for cartilage *in vitro*, and LDT administration produced synovial fluid concentrations of 1,353 ng/mL, remaining below concentrations considered potentially detrimental to cartilage. With HDT, synovial fluid total solids concentration was higher at 24 h and 7 days and sGAG concentration was higher at 48 h, compared to control joints. Synovial fluid CS-846 concentration was increased over pre-treatment values in HDT control but not in HDT treated joints at 24 and 48 h. All joints (HDT and LDT control and treated) showed a temporary decrease in synovial fluid C2C concentration, compared to pre-treatment values. Histologic features of articular cartilage and synovial membrane did not differ between HDT treated and control joints. High dose tiludronate treatment caused a transient increase in synovial total solids and temporarily increased proteoglycan degradation in cartilage. Although clinical significance of these changes are questionable, as they did not result in articular cartilage damage,

further investigation of the safety of intraarticular HDT in a larger number of horses is warranted.

## INTRODUCTION

Tiludronate, a non-nitrogen containing bisphosphonate, slows bone turnover by causing apoptosis in osteoclasts (*Rogers et al., 2011*). This effect may be useful to treat conditions associated with an increase in bone turnover, such as navicular disease or osteoarthritis. In fact, some horses with navicular disease (*Denoix, Thibaud & Riccio, 2003*), osteoarthritis of the distal tarsal joints (*Gough, Thibaud & Smith, 2010*) or the thoracolumbar facet joints (*Coudry et al., 2007*) exhibited decreased signs of pain associated with their condition after systemic tiludronate treatment. In humans, atypical subtrochanteric and femoral shaft fractures have been reported after years of treatment (*Nieves & Cosman, 2010*). In horses, only anecdotal reports of side effects are available to date, which have included colic, tachycardia, electrolyte imbalances, and presumed association with renal failure. Concerns regarding side effects and the high cost of systemic treatment constitute possible reasons why veterinarians have begun to administer tiludronate locally via intraarticular injection or regional limb perfusion. However, no information is available concerning safety, efficacy, or appropriate doses of these routes of administration. Based on *in-vitro* results (*Duesterdieck-Zellmer, Driscoll & Ott, 2012*), the extra-label use of tiludronate via intraarticular injection or regional limb perfusion, especially in high doses, may result in deleterious effects for articular cartilage. While it is common belief among veterinarians who have been using this drug in an extra-label fashion, that no untoward effects exist, this view is scientifically unfounded. Further, lameness after treatment is likely being attributed to the cause for the original lameness problem and not to negative effects of tiludronate. Thus, it is imperative to investigate possible deleterious effects of extra-label routes of administration for tiludronate in horses.

Little is known about tiludronate's effects on joint tissues after intraarticular administration. Concentration-dependent effects of tiludronate on articular cartilage *in-vitro* (*Duesterdieck-Zellmer, Driscoll & Ott, 2012*) raise concerns about the safety of higher doses administered intraarticularly. More specifically, tiludronate concentrations of $\geq 19{,}000$ ng/mL increased chondrocyte apoptosis and release of glycosaminoglycans from equine articular cartilage explants, suggesting that doses resulting in synovial fluid concentrations of $\geq 19{,}000$ ng/mL may promote articular cartilage damage (*Duesterdieck-Zellmer, Driscoll & Ott, 2012*). Anecdotally, the intraarticular dose of tiludronate is 50 mg per joint, which is likely to result in synovial fluid concentrations above 19,000 ng/mL when injected into the middle carpal joint, based on an estimated synovial fluid volume of 15 mL for this joint (*Ekman et al., 1981*). This dose was chosen as the high dose in this study, whereas the low dose (0.017 mg of tiludronate per joint) was chosen to result in synovial

fluid concentrations of about 1,900 ng/mL. The objective of this study was to determine effects of low dose tiludronate (LDT) and high dose tiludronate (HDT) administered intraarticularly to healthy horses. We hypothesized that a single intraarticular injection of HDT, but not of LDT would have a negative impact on clinical, clinicopathologic, and biochemical indicators of joint health, compared to observations in contralateral control joints injected with the same volume of saline. We further hypothesized that these negative effects would result in histologically detectable articular cartilage damage in joints administered HDT.

## MATERIALS AND METHODS

### Animals

After approval from the Institutional Animal Care and Use Committee, four horses (age 12–15 years, two Thoroughbreds, one Warmblood, one Quarter Horse) were enrolled to receive LDT (ACUP 4034, approved 6/29/10) and a year later, four different horses (age 2–18 years, Quarter Horses) were enrolled to receive HDT (ACUP 4160, approved 4/26/11). All horses were without front limb lameness and radiographic abnormalities of their carpal joints. They were housed in box stalls and offered free choice grass hay and water during experiments.

### Study design

Baseline lameness examinations were conducted, followed by sedation with detomidine (Dormosedan; Pfizer) and butorphanol (Torbugesic, Fort Dodge; both 0.01 mg/kg IV) for arthrocentesis of both middle carpal joints (performed by KDZ and LM). After withdrawing up to 2 mL joint fluid (used to determine all synovial fluid variables as described below), 0.017 mg tiludronate (Tildren; Ceva) in 1 mL saline was injected into one randomly assigned middle carpal joint of four horses, and 1 mL saline was injected into the contralateral middle carpal joint (year one, LDT). The other four horses (year two, HDT) were treated similarly, using an intraarticular dose of tiludronate of 50 mg in 6 mL saline as treatment and 6 mL saline as control. Randomization was performed via coin toss for each horse and investigators were blinded to treatment allocation by an assistant removing the label from the syringes before injection. Both joints were flexed and extended manually 15 times after injection. Aspiration of 0.5 mL joint fluid (used purely for determination of tiludronate concentrations) was performed after 10 min on both middle carpal joints. This resulted in removal of an estimated reduction of the effective intraarticular dose of tiludronate by 3% for LDT and 2.5% for HDT based on a presumed synovial fluid volume of 15 mL in the middle carpal joints (*Ekman et al., 1981*) and the assumption that none of the saline used as diluent for tiludronate had been absorbed from the joint space by 10 min post treatment. Repeat arthrocenteses of up to 3 mL joint fluid were performed after 24 and 48 h, 7 and 14 days (used for determination of all synovial fluid variables as described below). All horses treated with HDT were euthanatized on day 14 (Beuthanasia; Merck; 1 mL/4.54 kg IV) and articular cartilage from the radial facet of

both third carpal bones and adjacent synovial membrane were collected. Investigators were blinded to treatment allocations for all sample analyses.

## Clinical variables

Horses underwent daily physical and lameness examinations (performed by KDZ and LM), including palpation of both carpi for pain, effusion (none, mild, moderate, severe) and development of edema, measurement of joint circumference of the middle carpal joints at the level of a clipped mark on the lateral aspect of each middle carpal joint, and measurement of the angle of maximum flexion of both carpi (Baseline Plastic Goniometer; Fabrication Enterprises). For the latter, the angle present on the caudal/palmar aspect of the limb was measured. Thus, an increase in angle of maximum flexion would indicate decreased range of motion of the carpal joints. Horses were observed walking and trotting in a straight line, and trotting after flexion of both carpi for 1 min. Lameness was categorized as sound, intermittently lame at the walk or trot, consistently lame at the trot, or consistently lame at the walk. Responses to carpal flexion tests were graded as positive when obvious lameness persisted for more than four strides (*Ross, 2011*).

## Synovial fluid analyses

Synovial fluid samples were divided into 0.3 ml aliquots, of which one was placed in EDTA containing tubes (Monoject; Covidien) for cytologic analysis. This aliquot was stored at 4 °C until cytologic analysis within 12 h of sample collection. All other aliquots were stored at −80 °C until sample analyses within 6 months of collection. All samples from year one (LDT) were analyzed as one batch, and all samples from year two (HDT) were analyzed as one separate batch.

Total solids concentration was measured via refractometry (E-line Veterinary, Bellingham+Stanley). Total nucleated cell count was determined manually (Leuko-Tic, Bioanalytic). Differential cell counts were obtained manually on Wright's-Giemsa stained cytocentrifuge-prepared slides (CytoSpin*4 Cytocentrifuge; Thermo Scientific).

Synovial fluid sulfated glycosaminoglycan (sGAG) concentration was determined employing the 1,9-dimethylmethylene blue (DMMB) assay as described (*Oke et al., 2003*), with some modifications. Briefly, samples were added to equal volumes of papain digestion buffer (2 mg/mL papain) and mixed at 65 °C for 3 h, followed by an additional 1:3 (v:v) dilution in papain digestion buffer. Each sample was placed in triplicates into wells (5 μL/well) of a 96 well plate (nunc Optical Bottom Plate non-treated; Thermo Scientific). Immediately before measurement of absorbance at 540 nm, 245 μL of DMMB working solution (0.016 mg/mL DMMB) was added to each appropriate well using a multichannel pipette. Sample concentrations were calculated from a standard curve of chondroitin-6-sulfate (31.25–500 μg/mL; Chondroitin 6-sulfate from shark cartilage, Sigma Aldrich).

Aggrecan synthesis in articular cartilage was estimated by quantitating chondroitin sulfate 846 epitope (CS-846) in synovial fluid by ELISA (IBEX Technologies), according to manufacturer's instructions. Briefly, synovial fluid samples were centrifuged (16,000 × g for 2 min) and diluted 1:60 in Buffer III (provided by the manufacturer) and run in

duplicates. A color change was measured at 450 nm (Multiskan Go; Thermo Scientific) and sample concentrations were calculated from a standard curve of 20–1,000 ng/mL CS-846.

Collagen type II degradation was estimated by quantitating collagen type II cleavage neoepitope (C2C) in synovial fluid via ELISA (IBEX Technologies) according to manufacturer's instructions. Briefly, synovial fluid samples were centrifuged ($16,000 \times g$ for 2 min) and diluted 1:2 in Buffer III (provided by the manufacturer) were run in duplicates. A color change was measured at 450 nm (Multiskan Go; Thermo Scientific). Sample concentrations were calculated from a standard curve of 10–1,000 ng/mL C2C.

Tiludronate concentration was determined by high performance liquid chromatography (XBridge phenyl column; Waters) followed by mass spectrometry (*Tarcomnicu et al., 2009*; *Zhu et al., 2006*). Tiludronate was methylated using 0.2 M trimethylsilyldiazomethane in acetone. Sample concentrations were calculated from a standard curve of 0.5–64 ng/mL tiludronate. Samples with tiludronate concentrations above the range of the standard curve were serial diluted to fall within that range. Positive and negative control samples were run with each batch of samples and deuterated tiludronate (Toronto Research Chemicals) was used as internal control in each sample. The lower level of detection for this assay was 10 ng/mL.

## Joint tissue analyses ($n = 4$, HDT)

Cartilage samples from the radial facet of both 3rd carpal bones were digested with papain (1 mL papain digestion buffer as described above per 10 mg cartilage wet weight) for $\geq$24 h until no particulate matter was visible in the sample. Subsequently, sGAG content was determined via DMMB assay as described above.

Histologic analysis on paraffin embedded sections of cartilage and synovial membrane was performed by a board-certified veterinary pathologist (CVL) who was unaware of treatment allocations. Hematoxylin and eosin stained cartilage sections were graded on a scale of 0–4 for chondrocyte necrosis, chondrocyte clusters, cartilage fibrillation or fissures, and focal cell loss (*McIlwraith et al., 2010*). Cartilage sections stained with toluidine blue (for proteoglycans) were graded on a scale of 0–4 for loss of staining (*McIlwraith et al., 2010*). Hematoxylin and eosin stained synovial membrane sections were graded on a scale of 0–4 for cellular infiltration with lymphocytes and plasma cells, and intimal hyperplasia (*McIlwraith et al., 2010*). Scores were recorded for three independent readings and the median score was determined.

Chondrocyte apoptosis was assessed on cartilage sections using the terminal deoxynucleotidyl transferase dUTP nick end labeling (TUNEL) method (*in situ* Cell Death Detection Kit, AP, Roche) as described previously (*Duesterdieck-Zellmer, Driscoll & Ott, 2012*).

## Data analysis

Data were expressed as median and total range, other than tiludronate concentrations, which were expressed as mean and +95% confidence interval. To assess possible negative effects of intraarticular administration of LDT or HDT on normal joints, differences

between treated and control joints within the same horse were determined using a two factor factorial model (dose of tiludronate and treatment with saline or tiludronate) with repeated measures via non-parametric ANOVA. To assess effects of LDT or HDT on normal joints over time, differences between measurements prior to experimental treatment (baseline) and time points after experimental treatment within the same joints (LDT control joints, LDT treated joints, HDT control joints, or HDT treated joints) were determined. Comparisons between joints treated with HDT and joints treated with LDT were not performed, as experiments were conducted in two different years and samples were analyzed in different batches. Further, this comparison is of lower importance, as the study did not aim at determining which dose was safer than the other. For variables in which the main effect of time was not significant, median values for each treatment group (LDT control joints, LDT treated joints, HDT control joints, HDT treated joints) were calculated by pooling all time points. For those variables in which time significantly influenced the outcome, medians were calculated for each time point separately. Data from post-mortem analyses in horses treated with HDT was analyzed using the Wilcoxon signed-rank test. Statistical significance was set at $P = 0.05$.

A power analysis performed prior to the experiments estimated an 80% chance of detecting a difference of 30% between treated and control joints within the same horse with a sample size of $n = 4$ and $P < 0.05$.

## RESULTS

All physical examination variables remained within normal limits in all horses. No pain or edema was evident upon palpation of any carpi, and no lameness or positive response to carpal flexion was noted after tiludronate administration at any time. Mild to moderate effusion of the middle carpal joints was common 1–3 days and 8 days after intraarticular injection of saline, LDT and HDT (Table 1). Joints treated with HDT showed a greater degree of effusion than their saline controls 1 day ($P = 0.0162$), 3 days ($P = 0.0349$), and 8 days ($P = 0.0349$) after intraarticular injection. In contrast, in the LDT group, joints treated with saline showed a greater degree of effusion on day eight ($P = 0.0349$) than joints treated with LDT. When comparing joint effusion within experimental groups to baseline observations, HDT control joints showed increased effusion 2 days after injection of saline ($P = 0.0008$), whereas HDT treated joints showed increased effusion 2 days ($P = 0.0264$), 3 days ($P = 0.0264$) and 8 days ($P = 0.0264$) after treatment. LDT control joints, showed increased joint effusion 1 day ($P = 0.0264$), 2 days ($P < 0.0001$) and 8 days ($P = 0.0008$) after intraarticular injections, whereas LDT treated joints showed increased joint effusion 1 day ($P = 0.0264$) and 2 days ($P < 0.0001$) after treatment.

Joint circumference did not differ over time ($P = 0.7326$) nor between treated and control joints ($P = 0.3735$) in horses treated with HDT (28.3 [3.8] cm vs. 28.4 [2.7] cm— median and range pooled over time) or LDT (29.0 [5.9] cm vs. 28.8 [6.5] cm—median and range pooled over time).

Joint angle of maximum flexion was smaller in limbs treated with HDT than controls 10 ($P = 0.0135$) and 13 days ($P = 0.0219$) after treatment. There was no difference

**Table 1** Number of horses with middle carpal joint effusion before and after intraarticular administration of saline, LDT or HDT.

| Day of study | LDT control joints $n = 4$ | | LDT treated joints $n = 4$ | | HDT control joints $n = 4$ | | HDT treated joints $n = 4$ | |
| | Degree of joint effusion | | | | | | | |
| | Mild | Moderate | Mild | Moderate | Mild | Moderate | Mild | Moderate |
|---|---|---|---|---|---|---|---|---|
| 0[a] | 0 | 0 | 0 | 0 | 0 | 0 | 0 | 0 |
| 1[b] | 3 | 0 | 3 | 0 | 0 | 0 | 1 | 1 |
| 2[b] | 2 | 2 | 2 | 2 | 4 | 0 | 4 | 0 |
| 3 | 1 | 0 | 2 | 0 | 2 | 0 | 4 | 0 |
| 4 | 0 | 0 | 0 | 0 | 2 | 0 | 2 | 0 |
| 5 | 0 | 0 | 0 | 0 | 2 | 0 | 2 | 0 |
| 6 | 1 | 0 | 0 | 0 | 0 | 0 | 1 | 0 |
| 7[b] | 1 | 0 | 0 | 0 | 0 | 0 | 0 | 0 |
| 8 | 4 | 0 | 2 | 0 | 2 | 0 | 4 | 0 |
| 9 | 2 | 0 | 1 | 0 | 0 | 0 | 1 | 0 |
| 10 | 0 | 0 | 0 | 0 | 0 | 0 | 1 | 0 |
| 11 | 1 | 0 | 0 | 0 | 0 | 0 | 0 | 0 |
| 12 | 1 | 0 | 1 | 0 | 0 | 0 | 0 | 0 |
| 13 | 1 | 0 | 0 | 0 | 0 | 0 | 0 | 0 |
| 14[b] | 1 | 0 | 0 | 0 | 0 | 0 | 0 | 0 |

Notes.

[a] Observations were made prior to any intraarticular injections or arthrocenteses.

[b] Days with repeat arthrocenteses. Observations were made prior to that day's arthrocenteses.

between treated and control joints in horses treated with LDT (all $P > 0.3266$) at any time. Joint angle of maximum flexion was decreased at multiple time points compared to baseline measurements in HDT treated, HDT control, LDT treated, and LDT control joints (Table 2). All differences were <10°, and are unlikely to be clinically significant (*Liljebrink & Bergh, 2010*).

Synovial fluid concentrations of tiludronate are illustrated in Fig. 1. Tiludronate was detectable in all joints treated with HDT throughout the study, whereas in joints treated with LDT, tiludronate was not detectable after 24 h. Tiludronate was detectable in two HDT control joints 24 h after treatment, albeit below the lower margin of quantitation for the assay. Tiludronate was not detectable in any other samples from control joints or in any baseline samples.

Synovial total nucleated cell count did not differ between treated and control joints in horses treated with HDT or LDT ($P = 0.1165$). Total nucleated cell counts exceeded the reference interval of 0–500 cells/μL (*Mahaffey, 2002*) in three HDT treated joints, including one heavily blood contaminated sample, and in one HDT control joint 7 days after treatment. All other values remained within that reference range. When compared to total nucleated cell counts prior to intraarticular injection, cell counts were increased 24 h, 7 days and 14 days (all $P < 0.0001$) after treatment in HDT treated joints and 24 h, 48 h, 7 days and 14 days (all $P < 0.0042$) after treatment in HDT control joints. Total nucleated cell counts in samples from joints treated with LDT decreased 7 days after treatment compared

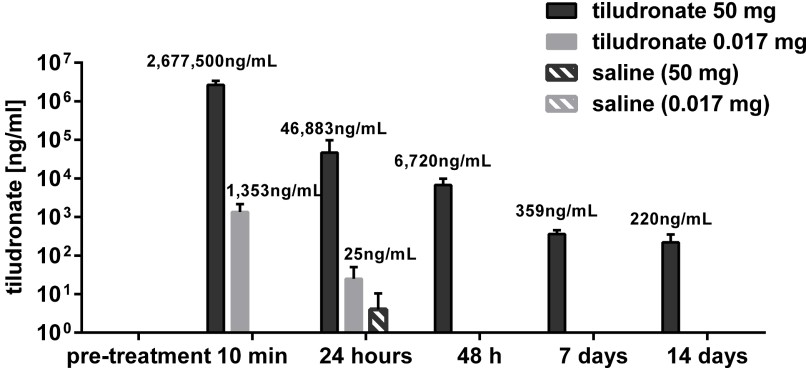

**Figure 1 Semi-log bar graph of mean synovial fluid tiludronate concentration over time.** Tiludronate was injected at a dose of 0.017 mg in 1 ml saline (A) or at a dose of 50 mg in 6 ml saline (B) into normal middle carpal joints of 4 horses, respectively. The contralateral joints were injected with an equal volume of saline to serve as control joints. Tiludronate concentratin was determined using HPLC followed by mass spectrometry. Error bars show +95% confidence interval of measurements from 4 horses.

**Table 2 Median joint angle of maximum flexion and total range in horses treated with either LDT or HDT in one middle carpal joint and with saline in the contralateral joint.**

| | LDT control joints | | LDT treated joints | | HDT control joints | | HDT treated joints | |
|---|---|---|---|---|---|---|---|---|
| Day of study | Median | Total range | Median | Total range | Median | Total range | Median | Total range |
| 0[a] | 11 | 6 | 13 | 8 | 4 | 6 | 3.5 | 3 |
| 1[b] | 8.5* | 3 | 9* | 4 | 3 | 2 | 4.5 | 6 |
| 2[b] | 10.5 | 6 | 8* | 4 | 2.5 | 3 | 2 | 1 |
| 3 | 6.5* | 2 | 7* | 6 | 2* | 1 | 2 | 2 |
| 4 | 9* | 4 | 7.5* | 3 | 2* | 2 | 2* | 2 |
| 5 | 8* | 1 | 8* | 3 | 2* | 1 | 2 | 1 |
| 6 | 8* | 7 | 8* | 5 | 1* | 2 | 1* | 3 |
| 7[b] | 11 | 6 | 11 | 3 | 2* | 4 | 2.5 | 6 |
| 8 | 14 | 7 | 12.5 | 3 | 2* | 1 | 2.5 | 3 |
| 9 | 11.5 | 3 | 10 | 4 | 0* | 2 | 2 | 3 |
| 10 | 11.5 | 3 | 12 | 6 | 0.5*,** | 2 | 3** | 3 |
| 11 | 11.5 | 4 | 11.5 | 2 | 2* | 3 | 2 | 0 |
| 12 | 9 | 6 | 9* | 4 | 1* | 2 | 1.5 | 3 |
| 13 | 7* | 3 | 7* | 2 | 2*,** | 2 | 3.5** | 3 |
| 14[b] | 8* | 4 | 9* | 2 | 0.5* | 2 | 1* | 2 |

**Notes.**

[a] Measurements were taken prior to any intraarticular injections or arthrocenteses.

[b] Days with repeat arthrocenteses. Measurements were taken prior to that day's arthrocenteses.

* Significantly different from baseline measurement on day 0 (*$P < 0.05$). However, differences in angulation of $<10°$ are unlikely to be clinically significant (*Liljebrink & Bergh, 2010*).

** Significant difference between HDT treated and control joints (**$P < 0.022$). However, differences in angulation of $<10°$ are unlikely to be clinically significant (*Liljebrink & Bergh, 2010*).

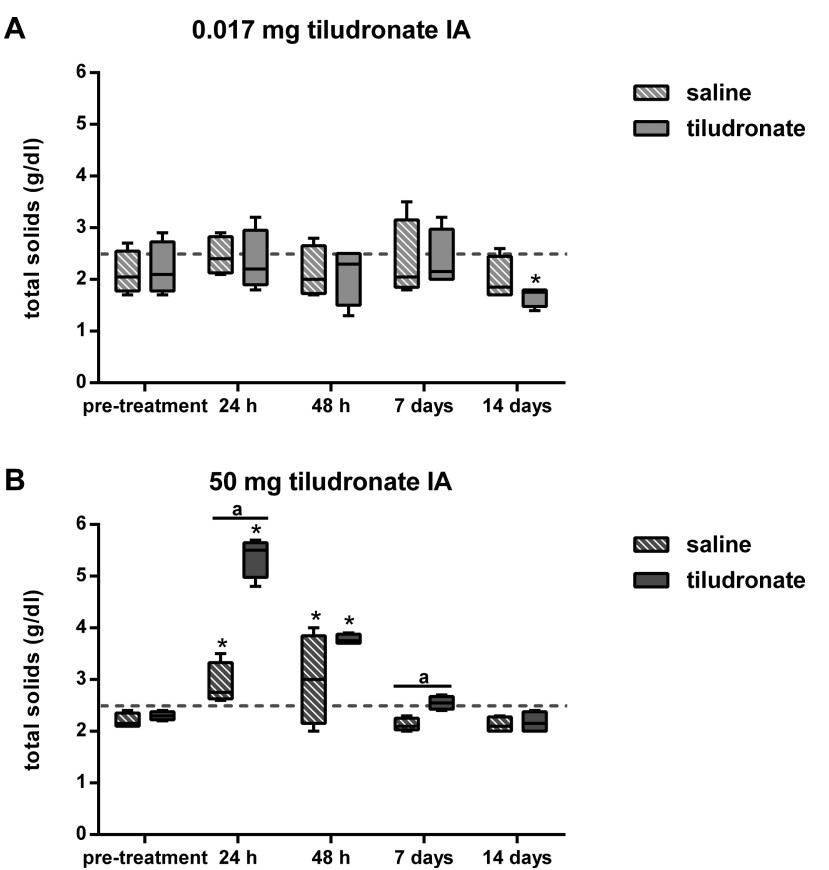

**Figure 2** **Box-and-whisker plot of synovial fluid total solids concentration over time.** Tiludronate was injected at a dose of 0.017 mg in 1 ml saline (A) or at a dose of 50 mg in 6 ml saline (B) into normal middle carpal joints of 4 horses, respectively. The contralateral joints were injected with an equal volume of saline to serve as control joints. Total solids concentration was determined via refractometry. The dotted horizontal line represents the upper limit of the normal reference interval (*Davidson & Orsini, 2007*). Whiskers represent the minimum and maximum values. Asterisks indicate significant difference compared to pre-treatment measurements in the same joints (*$P < 0.05$). The superscript "a" indicates significant difference between treated and control joints ($P < 0.05$).

to baseline values ($P = 0.0124$), and no change over time was found for LDT control joints (all $P > 0.7$).

The percentage of neutrophils among nucleated cells in synovial fluid samples remained within the reference interval of $\leq 10\%$ (*Mahaffey, 2002*) in the majority of samples (79%). The remaining samples had 12–68% neutrophils, which was considered to be within normal limits due to the very low total nucleated cell counts in these samples (*Mahaffey, 2002*). The percentage of neutrophils among nucleated cells in synovial fluid samples did not differ over time ($P = 0.0871$) nor between treated and control joints ($P = 0.3386$) in horses treated with HDT (4.5 [68]% vs. 5.5 [55]%—median and range pooled over time) or LDT (2 [20]% vs. 5 [25]%—median and range pooled over time).

Total solids concentration in synovial fluid (Fig. 2) was higher in HDT treated joints than HDT control joints but there was no difference between LDT treated and LDT control joints. Compared to baseline values, total solids increased in HDT treated and

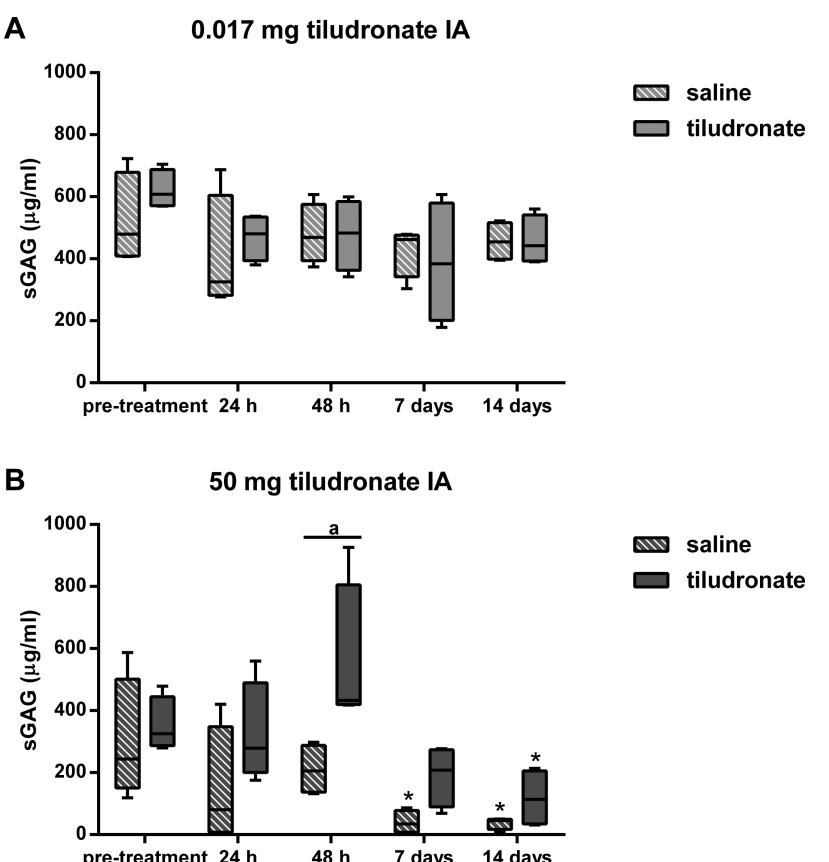

**Figure 3 Box-and-whisker plot of synovial fluid sulfated glycosaminoglycan (sGAG) concentration over time.** Tiludronate was injected at a dose of 0.017 mg in 1 ml saline (A) or at a dose of 50 mg in 6 ml saline (B) into normal middle carpal joints of 4 horses, respectively. The contralateral joints were injected with an equal volume of saline to serve as control joints. Sulfated glycosaminoglycans were quantitated using the 1,9-dimethylmethylene blue (DMMB) assay. Whiskers represent the minimum and maximum values. Asterisks indicate significant difference compared to pre-treatment measurements in the same joints (*$P < 0.05$). The superscript "a" indicates significant difference between treated and control joints ($P < 0.05$).

HDT control joints for the first 2 days. In LDT treated joints, total solids concentration was decreased at the end of the study, compared to baseline values and there was no difference compared to baseline in LDT control joints.

Synovial fluid sGAG concentration (Fig. 3) was greater in HDT treated than HDT control joints on day 2 of the study, but there was no difference between LDT treated and LDT control joints. Compared to baseline values, sGAG concentration tended to increase ($P = 0.0513$) and then decreased over time in HDT treated joints, whereas no increase compared to baseline was seen in the HDT control joints, and instead, values decreased compared to baseline. In LDT treated or control joints, sGAG concentration did not change compared to baseline.

Synovial fluid CS-846 concentration (Fig. 4) was lower in HDT treated joints compared to HDT control joints, whereas there were no differences between LDT treated and LDT control joints. In HDT control joints, synovial fluid CS-846 concentration increased and

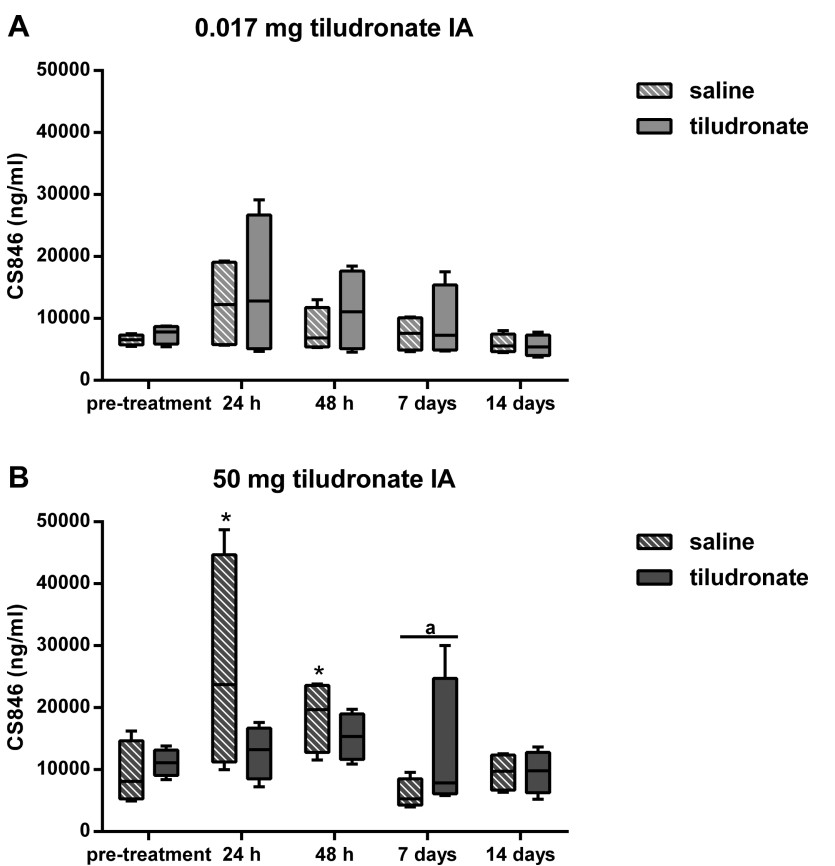

**Figure 4** **Box-and-whisker plot of synovial fluid concentration of chondroitin sulfate 846 epitope (CS-846) over time.** Tiludronate was injected at a dose of 0.017 mg in 1 ml saline (A) or at a dose of 50 mg in 6 ml saline (B) into normal middle carpal joints of 4 horses, respectively. The contralateral joints were injected with an equal volume of saline to serve as control joints. Chondroitin sulfate 846 epitope was quantitated using ELISA to estimate aggrecan synthesis. Whiskers represent the minimum and maximum values. Asterisks indicate significant difference compared to pre-treatment measurements in the same joints (*$P < 0.05$). The superscript "a" indicates significant difference between treated and control joints ($P < 0.05$).

then returned to not different from baseline levels again and this was not seen in HDT treated joints, as there was no change compared to baseline. In LDT treated and LDT control joints, no change in synovial fluid CS-846 concentration compared to baseline was found.

Synovial fluid C2C concentration (Fig. 5) was lower in HDT treated than control joints, as well as in LDT treated compared to LDT control joints. All joints, HDT treated and HDT control joints as well as LDT treated and LDT control joints showed a temporary decrease in C2C concentration, compared to baseline values.

No significant difference was found between joints treated with HDT and control joints for sGAG content of articular cartilage (308.0 [15.9] μg/mg vs. 308.6 [46.8] μg/mg respectively; $P = 1.0$) or % of apoptotic chondrocytes (2.2 [1.2]% vs. 1.7 [0.6]%; $P = 0.5637$). Histologic scores for joint tissues were not different between joints treated with HDT and control joints (Table 3).

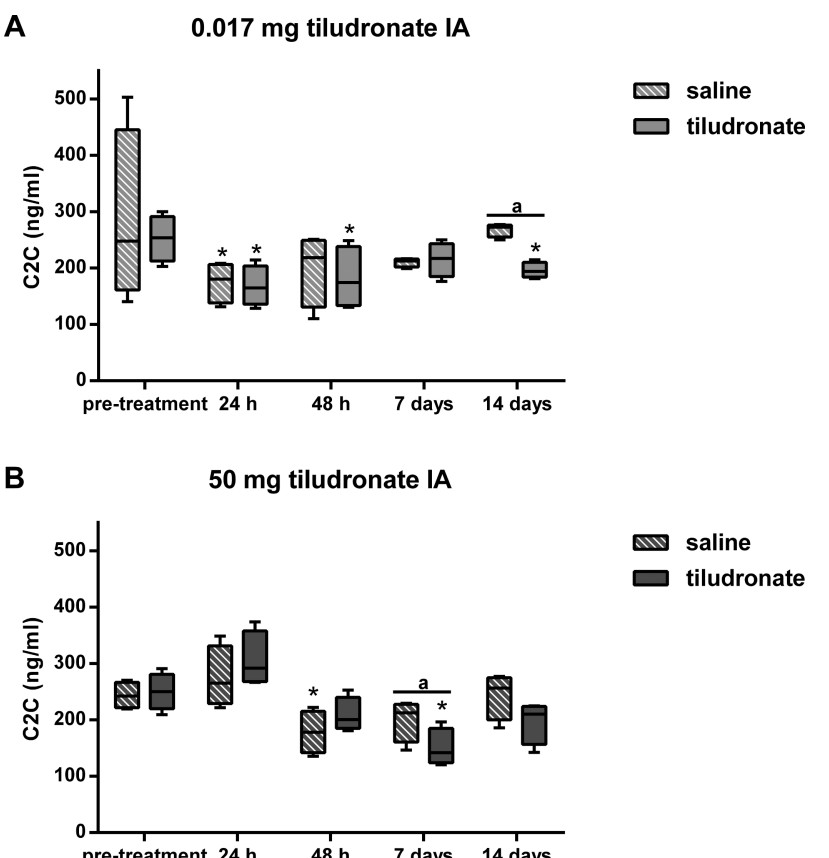

**Figure 5 Box-and-whisker plot of synovial fluid concentration of collagen type II cleavage neoepitope (C2C) over time.** Tiludronate was injected at a dose of 0.017 mg in 1 ml saline (A) or at a dose of 50 mg in 6 ml saline (B) into normal middle carpal joints of 4 horses, respectively. The contralateral joints were injected with an equal volume of saline to serve as control joints. Collagen type II cleavage neoepitope was quantitated using ELISA to estimate collagen type II degradation. Whiskers represent the minimum and maximum values. Asterisks indicate significant difference compared to pre-treatment measurements in the same joints (*$P < 0.05$). The superscript "a" indicates significant difference between treated and control joints ($P < 0.05$).

## DISCUSSION

The target tissue for tiludronate has traditionally been bone (*EMEA, 2001*), but tiludronate also exerts direct effects on cartilage and chondrocytes (*Duesterdieck-Zellmer, Driscoll & Ott, 2012*; *Emonds-Alt, Breliere & Roncucci, 1985*), especially after extra-label local administration via regional limb perfusion or intraarticular injection, since these routes of administration potentially result in high synovial fluid concentrations. *In-vitro* data suggests that synovial fluid tiludronate concentrations of 19,000 ng/mL and above promote chondrocyte apoptosis and release of sGAGs from articular cartilage matrix, whereas concentrations of 1,900 ng/mL and below may ameliorate sGAG release and chondrocyte apoptosis in equine joints (*Duesterdieck-Zellmer, Driscoll & Ott, 2012*). Subsequently, the dose for the LDT group was chosen to produce synovial fluid tiludronate concentrations approximating 1,900 ng/mL. The dose for HDT was selected based on

**PeerJ** _______________

**Table 3 Median histologic scores of articular cartilage and synovial membrane 14 days after intraarticular injection of 50 mg tiludronate or saline.**

| Histologic variable | Treatment | Median score[a] | Total range | *P*-value |
|---|---|---|---|---|
| Chondrocyte necrosis | Tiludronate | 0 | 0 | N/A |
| | Saline | 0 | 0 | |
| Cluster formation | Tiludronate | 1 | 1 | >0.999 |
| | Saline | 1 | 1 | |
| Fibrillation/fissuring | Tiludronate | 0.5 | 1 | >0.999 |
| | Saline | 1 | 1 | |
| Focal chondrocyte loss | Tiludronate | 0 | 0 | N/A |
| | Saline | 0 | 0 | |
| Cartilage toluidine blue stain uptake | Tiludronate | 1.5 | 1 | >0.999 |
| | Saline | 1 | 2 | |
| Synovial cellular infiltration (lymphocytes and plasma cells) | Tiludronate | 1 | 2 | 0.25 |
| | Saline | 0 | 1 | |
| Synovial intimal hyperplasia | Tiludronate | 0.5 | 1 | >0.999 |
| | Saline | 0.5 | 1 | |

**Notes.**

[a] Histology was scored on a scale of 0–4, with 0 representing normal histology (*McIlwraith et al., 2010*).

anecdotal reports of clinical use and was suspected to generate synovial fluid tiludronate concentrations exceeding those considered safe for articular cartilage. As suspected, HDT resulted in synovial fluid concentrations much higher than 1,900 ng/mL for at least 48 h after intraarticular injection. This was accompanied by increased synovial fluid sGAG concentration, resulting either from increased degradation and/or increased turnover of sGAGs in articular cartilage (*de Grauw, van de Lest & van Weeren, 2009*), which would be associated with an increased sGAG synthesis. To estimate synthesis of sGAGs, synovial CS-846 concentrations were determined, as this epitope has been used as a biomarker for aggrecan synthesis (*McIlwraith, 2005*; *Poole et al., 1994*; *Rizkalla et al., 1992*). Aggrecan synthesis appeared to remain unchanged in joints treated with HDT compared to measurements prior to treatment. Thus, increased synovial sGAG concentrations in joints treated with HDT in the present study were most likely associated with cartilage matrix degradation and not with an increased turnover of sGAGs. Increased concentration of sGAG in synovial fluid has also been reported due to repeated arthrocentesis in horses (*Van den Boom et al., 2005*). However, it is unlikely that the increase in sGAG concentration 48 h after intraarticular injection of HDT was caused by repeated arthrocenteses, as sGAG concentration in contralateral control joints that had undergone the same repeat arthrocentesis protocol as the treated joints were significantly lower than in tiludronate treated joints. Further, repeat arthrocentesis caused a more modest increase in synovial fluid sGAG concentration of about 20 μg/mL (*Van den Boom et al., 2005*) compared to what was found after HDT in this study (increase in synovial fluid concentratin of about 200 μg/mL). Nevertheless, sGAG content in articular cartilage was not found to be different between joints treated with HDT and control joints 14 days after treatment, though this may reflect the small number of horses investigated. A power analysis performed prior to

the experiments estimated an 80% chance of detecting a difference of 30% between treated and control joints within the same horse with a sample size of $n = 4$ and $P < 0.05$. Another explanation for the lack of difference in sGAG content between HDT treated and control joints may be that tiludronate concentration in HDT treated joints remained within a range that has been shown to decrease sGAG release from articular cartilage matrix *in vitro* (*Duesterdieck-Zellmer, Driscoll & Ott, 2012*) for at least the last seven days of the study. This may have allowed for increased retention of sGAGs, thus ameliorating earlier sGAG losses.

Similar to what was observed in HDT control joints in this study, aggrecan synthesis, measured as CS-846 concentration in synovial fluid, has been reported to increase with repeated arthrocentesis after 24 and 48 h in horses (*Lamprecht & Williams, 2012*). This increase was ameliorated in joints injected with HDT in our study, suggesting that high synovial fluid concentrations of tiludronate may prevent chondrocytes anabolic response to mild insults such as repeated arthrocenteses. Exercise has also been shown to increase aggrecan synthesis by chondrocytes (*Lamprecht & Williams, 2012*), but it is impossible to state if intraarticular injection of HDT would ameliorate this response in a similar fashion.

Another major protein of articular cartilage matrix is collagen type II (*Vachon et al., 1990*), and it is accepted that collagen type II degradation in cartilage is irreversible (*Catterall et al., 2010*; *Jubb & Fell, 1980*). The impact of repeat arthrocenteses on synovial fluid C2C concentration appears to be variable over time in young horses (*Lucia et al., 2013*). After multiple arthrocenteses over a time period of 12 h, synovial fluid C2C concentration was increased at 24 h, decreased at 7 days and unchanged at 14 days compared to baseline values. In the present study, no increase in C2C was observed in LDT or HDT treated or control joints. Similarly to what was reported in young horses (*Lucia et al., 2013*), a decrease in C2C was seen in all experimental groups between 24 h and 14 days after treatment. Thus, changes in type II collagen cleavage are likely attributable to the effect of repeat arthrocenteses, as opposed to tiludronate treatment, although these results should be viewed in light of the small sample number used in this study.

Synovial fluid total solids concentration increased at 24 and 48 h in both treated and control joints of HDT horses. While this was likely due to effects from repeated arthrocenteses (*Francoz, Desrochers & Latouche, 2007*; *Jacobsen, Thomsen & Nanni, 2006*) and saline injections (*Wagner, McIlwraith & Martin, 1982*) for the control joints, total solids concentration increased to a much greater extent in HDT treated joints than in control joints. Thus, intraarticular administration of HDT was associated with a significant increase in synovial fluid total solids concentration, most likely due to increased synovial total protein concentration. Increased synovial protein concentration in HDT treated joints was attributed to be most likely caused by synovial inflammation, as these joints were also palpable effused 24 and 48 h after treatment, and in clinical cases, this observation is usually associated with a synovial inflammatory response (*Mahaffey, 2002*). However, as synovial fluid prostaglandin $E_2$ concentration was not determined in this study, we are unable to prove this assumption. It is unlikely that the high tiludronate concentration in samples from HDT treated joints increased the total solids readings on the refractometer, as spiking of blank synovial fluid samples with similar concentrations of tiludronate did

not change total solids readings (data not shown). Finally, while an increase in sGAG concentration can affect total solids readings on the refractometer (data not shown), the increased total solids concentration 24 h after HDT treatment cannot be explained with this phenomenon, as sGAG concentratin was not elevated at that time point.

An unexpected finding was a significant elevation of the total nucleated cell count in both, HDT treated and HDT control joints 7 days after treatment. This is unlikely to be due to repeat arthrocenteses, as no arthrocenteses had been performed for the previous 5 days and elevation of nucleated cell counts in synovial fluid due to repeat arthrocenteses have been reported to occur within 24–48 h after initial arthrocentesis in horses (*White et al., 1989*) and calves (*Francoz, Desrochers & Latouche, 2007*) and it has been suggested that joints can adapt to repeated arthrocenteses over time, resulting in fewer alterations of total protein concentration and nucleated cell counts after 24 h (*Francoz, Desrochers & Latouche, 2007*; *White et al., 1989*). Further, there was no increase in total nucleated cell count 7 days after treatment in horses treated with LDT. Thus, the cause for this increase in total nucleated cell count is unclear.

Although LDT resulted in synovial tiludronate concentrations that remained within a range shown to be safe and beneficial for articular cartilage *in vitro*, these concentrations were maintained only for 24 to 48 h, which may not be sustained enough to provide lasting beneficial effects. However, it is possible that this dose is sufficient to affect subchondral bone remodeling in treated joints if tiludronate diffuses through hyaline and calcified cartilage into adjacent subchondral bone, although it is unknown whether or not this occurs. Investigations to determine tiludronate content in subchondral bone after intraarticular administration are necessary to answer this question.

Interestingly, tiludronate was detectable in two control joints of horses treated with HDT, although concentrations were below the linear part of the standard curve for the tiludronate assay. This finding is most likely explained by absorption of tiludronate from synovial fluid by synovial capillaries, followed by redistribution into peripheral tissues, including distant joints. In support of this, radioactivity from technetium 99 medronate, a radioactively labeled bisphosphonate, was detectable in peripheral plasma within 5 min after intraarticular injection, peaked at about 45 min, and was still detectable 24 h post intraarticular injection (*Dulin et al., 2012*). Alternatively, contamination of the two samples from HDT control joints may have occurred during sample processing.

The first hypothesis, that a single intraarticular injection of HDT, but not of LDT would have a negative impact on variables of joint health, compared to contralateral control joints was statistically confirmed in this preliminary investigation. However, the observed changes are of questionable clinical significance and further investigation with a greater number of horses is warranted. The second hypothesis that intraarticular injection of HDT would result in histologically detectable articular cartilage damage was rejected, but the low power of this preliminary study has to be taken into consideration when interpreting this result.

## CONCLUSIONS

This study suggests that intraarticular administration of a high dose of tiludronate (50 mg) may result in temporary elevation of synovial fluid total solids and sGAG concentration. Further, this dose may prevent chondrocytes' short-term anabolic response to mild insults such as repeated arthrocenteses. However, a high dose of tiludronate did not appear to negatively affect cartilage or synovial membrane as assessed via histologic analysis 2 weeks after treatment. Thus, the clinical significance of these findings remains questionable and further study of possible negative effects of a high dose of tiludronate administered intraarticularly in a larger number of horses is warranted.

Intraarticular administration of a low dose of tiludronate (0.017 mg) did not appear to impact assessed variables of joint health, although this finding has to be interpreted with caution, due to the small number of horses investigated.

## ACKNOWLEDGEMENTS

The authors thank Drs. Stacy Semevolos and Erica McKenzie for their critical review of this manuscript.

### Funding

Funding for this study was provided by the Department of Clinical Sciences, College of Veterinary Medicine, Oregon State University, and the Knapp Friesian Foundation, Inc., in addition to the Merial Scholars Program and the Department of Clinical Sciences Summer Research Program of Oregon State University (L Moneta). Oregon State University's mass spectrometry facility is in part supported by the National Institute of Environmental Health Sciences (Grant Number P30ES000210). The funders had no role in study design, data collection and analysis, decision to publish, or preparation of the manuscript.

### Grant Disclosures

The following grant information was disclosed by the authors:
Department of Clinical Sciences, College of Veterinary Medicine, Oregon State University.
The Knapp Friesian Foundation, Inc.
Department of Clinical Sciences Summer Research Program of Oregon State University.
National Institute of Environmental Health Sciences: P30ES000210.

### Competing Interests

The authors declare there are no competing interests.

### Author Contributions

- Katja F. Duesterdieck-Zellmer conceived and designed the experiments, performed the experiments, analyzed the data, contributed reagents/materials/analysis tools, wrote the paper, prepared figures and/or tables, reviewed drafts of the paper, secured funding for the experiments.

- Lindsey Moneta and Maureen K. Larson conceived and designed the experiments, performed the experiments, analyzed the data, contributed reagents/materials/analysis tools, reviewed drafts of the paper.
- Jesse F. Ott conceived and designed the experiments, performed the experiments, contributed reagents/materials/analysis tools, reviewed drafts of the paper.
- Elena M. Gorman conceived and designed the experiments, contributed reagents/materials/analysis tools, reviewed drafts of the paper.
- Barbara Hunter and Claudia S. Maier performed the experiments, contributed reagents/materials/analysis tools, reviewed drafts of the paper.
- Christiane V. Löhr contributed reagents/materials/analysis tools, reviewed drafts of the paper.
- Mark E. Payton conceived and designed the experiments, analyzed the data, contributed reagents/materials/analysis tools, reviewed drafts of the paper.
- Jeffrey T. Morré performed the experiments, analyzed the data, contributed reagents/materials/analysis tools, reviewed drafts of the paper.

### Animal Ethics

The following information was supplied relating to ethical approvals (i.e., approving body and any reference numbers):

Institutional Animal Care and Use Committee of Oregon State University
ACUP approval numbers:
ACUP 4034, approved 6/29/10
ACUP 4160, approved 4/26/11.

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
