# Peer review of "Effects of low and high dose intraarticular tiludronate on synovial fluid and clinical variables in healthy horses—a preliminary investigation"

_PeerJ, doi:10.7717/peerj.534_

## Round 0.1 · original submission · Major Revisions

· Academic Editor

Major Revisions

The reviewers were quite divided as to their opinion of the suitability of this manuscript for publication. Reviewer #1 had a number of important comments that will need to be given serious consideration, and it likely that this reviewer will need to weigh in on any revision. In order for your manuscript to be acceptable for publication you will need to respond to all comments by both reviewers and find appropriate ways to address their criticisms in your rebuttal letter.

Reviewer 1 ·

Basic reporting

The effects of repeated arthrocentesis have been well studied by other investigators and is highly relevant to the interpretation of your data:
(1) van den Boom R, et al, Equine Vet J 2005;37:250
(2) Francoz D, Can J Vet Res 2007;71:129
Some discussion is needed on the possible effects of repeated arthrocentesis on your study.

Experimental design

Major concerns:
Although this in vivo study is a logical progression for the authors’ previous in vitro work on tiludronate, the study contains some major design flaws. Because of these flaws (described below), it is not possible to test the main hypothesis. The power calculation to determine the number of horses needed is not well described, but seems to be based on incorrect assumptions. Interpretation of the data, as presented, is unconvincing and possibly inaccurate. The Results section is laborious to read and would need to be rewritten in a more concise and clear fashion. Some statements seem to be contradictory as currently written.

Validity of the findings

The reviewer has concerns regarding the validity of the findings because of the design flaws, concerns about statistical analysis, and disregard for the effects of repeated arthrocentesis.

Additional comments

Line 44-49: The hypothesis statement reads very awkwardly. The points made parenthetically should be made before your hypothesis statement.

Line 59-61: The power calculation should go in your data analysis section. In what measurement or measurements were you expecting a 30% difference? Apparently not between the HDT and LDT groups, because they could not be compared.

Lines 63-70: The study design raises some concerns. The LDT and HDT groups had different volumes injected, which is probably not of consequence, but same volume could easily have been used to eliminate this difference. More concerning is the 1 year separation between groups. Why were the LDT and HDT groups not mixed in year 1 and year 2? This would have eliminated the potential effect of the 2 experiments being conducted under possibly different or very different conditions. This also applies to doing the synovial fluid analyses at different times and certainly different lot numbers for your ELISA kits. Manufacturers of these kits frequently make changes in the kits that are not always communicated to the user, and thus kits manufactured a year apart can cause variability in results.

Line 79: should read “…were collected.”

Line 83-84: What does “soft tissue swelling” mean? This seems very subjective. Surely measurement of joint circumference would be far more objective.

Line 88-89: Why would you not use the accepted AAEP grading scale? If that is what you are doing, you need to indicate that.

Line 104-105: This sentence is unclear. Was the sample again diluted in papain digestion buffer (presumably containing papain) after the digestion?

Line 109-110: What was the source of chondroitin-6-sulfate standard?

Lines 130: Did you calculate the percentage of the tiludronate dosage removed from the joint in the 10 minute sample? This would be very simple to do, right?

Line 141: It would be helpful to indicate what the staining is for.

Line 149: Use of 95% confidence interval would give a more critical evaluation of your data than SEM.

Line 152-154: You did no comparisons between groups and yet in your power analysis statement (line 59-61), you state you are expecting a difference of 30% between groups. How can you reconcile this? It seems that during the design phase of the experiment it should have been obvious that you were conducting 2 separate experiments, not one. Thus, you cannot test your hypothesis as stated. This was an unfortunate oversight that greatly weakens the impact of your study, which is very relevant to clinical practice.

Line 164-165: “Mild to moderate effusion of the middle carpal joints was common 1-3 days (50-100% of all joints)…” This is vague.

Line 171-172: What does “…for both groups” mean? Are you talking about LDT and controls as 2 different groups? This is unclear.

Line 173-177: The last 2 sentences of this paragraph seem to be the take home message of the paragraph. Everything in the paragraph preceding these 2 sentences is very confusing.

Line 181-188: The first sentence of this paragraph was straightforward, but the subsequent sentences are very difficult to follow.

Line 276-295: Although increased total solids in the synovial fluid is compatible with inflammation, it is not very much more than suggestive. Other factors besides inflammation can increase total solids. Previous in vitro work by the authors showed that tiludronate concentration had no effect on PGE2 in the culture medium or IL-6 expression. In absence of any more specific markers of inflammation, interpreting the response as inflammatory may be correct or incorrect. Although DMMB is commonly used and is an accepted assay to give some estimate of sGAG release, it is a very crude assay at best. That, combined with your small numbers and having 2 experiments separated by a year, raise concerns about the meaningfulness of your sGAG data. The relatively small increase in sGAG in 4 HDT joints at 48 hours, although statistically significant, is of questionable clinical importance. It might mean something if you had more horses, but probably not. A 95% CI would help in this judgment.

Line 308-310: Did you save subchondral bone samples suitable for tiludronate assay?

Line 320-321: What were the short-term negative effects? There was no histological evidence of this. Elevations of synovial fluid nucleated cell count at 7 days were basically within normal limits. Elevation of total solids at 24 and 48 hours might be the most remarkable finding, but of questionable significance. Can this be interpreted as a negative effect? Is it really evidence of an inflammatory response? Maybe, maybe not. sGAG was elevated only at 48 hours, but still less than pretreatment values in the LDT, so is that a negative effect? CS846 and C2C are also not very precise biomarkers and although you report a couple of significant differences, what is the clinical significance of these relatively small changes? Interpretation of your results as negative may be overstating your case. If anything, have you shown in your small number of horses that IA tiludronate did not cause any harm? Either way, as presented, your data is inconclusive.

Table 1: This information would be better presented graphically in a semi-log plot.

Figure 1, 2, 3, 4, & 5: These figures are unnecessary because they do not add to the description in results and greatly increase the length of the manuscript.

Reviewer 2 ·

Basic reporting

No comments

Experimental design

no comments

Validity of the findings

no comments

Additional comments

This is an interesting and important piece of basic evaluation on a clinically, extra-label used compound. The information obtained -though they are limited on short term evaluation and evaluated in a small group only- should be considered when using tiludronate intraarticularly. Inclusion of the subchondral bone into the evaluation would have potentially increased the results spectrum, knowing that osteoclast are the target cells of tiludronate. Little is known about treatment effects of tiludronate for osteoarthritis (OA), as with the treatment of OA overall. There seem to be various phenotypes of OA some of which are dominated by subchondral bone pathology. Thus, there might be an indication for this treatment approach but a lot more research is needed before this can be recommended safely. Based on the presented data, potential advantageous effects of a high vs. low dose tiludronate should be critically balanced against the here presented side effects on each individual case.

Line 46
“… 10,000 ng/mL”
different numbers compared to the discussion (see comment below)

Line 85
“measurement of angle”
What was measured? Is an increasing angle (results Line 183) a good or a bad result?

Line 149, also Table 1 and 2, Fig
“mean vs. median”
Why and when was one or the other chosen? Test for normal distribution? Little test groups (n=4) should always be represented by median, not by mean.

Line 178-180
“….. did not differ over time …. “
What are those given mean+/SEM data? On a certain time point or an average over time?

Line 181-183
“ … at any time”
See comment above. Same problem.

Line 201, Fig. 1
Any explanation for the significant result in the control group?

Line 219
incomplete brackets

Line 242, Fig 4
Any explanation for the significant result in the control group?

Line 272, 275
“1,900 ng/ml”
check and compare data to Line 46

Line 303
“re sulted”
delete space character

Line 313 and 317
“unlikely”
I was surprised that you refute systemic effect and rather consider contamination?
When and how was contamination possible? Are there data available of tiludronate concentration in the blood after systemic application?

Line 318
Please comment on your hypotheses

Figures in general
Use different colours or use shading instead: in a black and white copy, the gray values of the bars are hard to discern.

---

## Round 0.2 · Minor Revisions

· Academic Editor

Minor Revisions

Thank you for clarifying the manuscript's hypotheses and addressing the respective reviewers' concerns, particularly about not comparing the two treatment groups sampled a year apart.

In addition to the new comments from one of the reviewers, I want to respond to one of your rebuttal statements: "Our statistician suggested using means for datasets that subjectively to him (and also based on the Kolmorgorov-Smirnoff normality test, although it is not very powerful for small sample sets) were normally distributed and to use medians for datasets with outliers, or non-normal distribution based on the K-S test (joint effusion data, all histologic scores). We were unable to confirm in the literature that medians should be used for small test groups." There is no way with a sample size of four to convincingly demonstrate that data arose from a population of normally distributed values. In fact, it's easy to show that strongly non-normal data will yield a non-significant result on a normality test (e.g., Kolmorgorov-Smirnoff or Shaprio-Wilk); such tests do not have power with a sample size of four to detect non-normality. Therefore, although you can present your results as means and standard deviations, it is also necessary to present such data without assuming an underlying parametric (normal) distribution using medians and minimums/maximums. Wherever possible you should use nonparametric tests that do not assume normality; these are entirely appropriate for small test groups, and can yield exact p-values from hypothesis tests if done appropriately. In addition, if you are using repeated measures ANOVA, it is necessary to evaluate the normality of the residuals to confirm that the model is reasonable to use (even here there are nonparametric alternatives). I suggest you consult further with your statistician on accomplishing this.

Reviewer 2 ·

Basic reporting

no comment

Experimental design

no comment

Validity of the findings

no comment

Additional comments

space characters missing in Line 50, 306, 348

Line 58/59
doubling of the information “age 2-18 years, Quarter Horses”

Line 68, 151 (example)
consistency for numbers 0-9 as numerals or words throughout the manuscript

Line 91, 92; Table 2
So an increase in angle means a decreased range of motion, which one might expected with increased joint effusion. Then I don´t understand the numbers in Table 2: Why do (almost) all mean joint angles decrease after time point 0? Also, isn´t the angle in normal horses always 0°, i.e. the skin of the caudal aspect of the radius can be brought in contact with the palmar aspect of the metacarpus? And does this not also depend on the force applied, i.e. how much I squeeze those 2 parts of the limb together?

Line 134
source of tiludronate was provided already in line 67

Line 156
consistency of the order using ( ) vs. [ ] brackets

Line 293
closing brackets

line 367
„AtAlternatively“

Table 1
- maybe add “middle carpal”
- maybe include a comment on how many animals were included per group, otherwise a number like “3” animals is hard to put in perspective

---

## Round 0.3 · accepted · Accept

· Academic Editor

Accept

I believe the reconsidered analyses strengthen your paper's overall validity.